# Investigating Perceptions of Teachers and School Nurses on Child and Adolescent Oral Health in Los Angeles County

**DOI:** 10.3390/ijerph19084722

**Published:** 2022-04-14

**Authors:** Carl A. Maida, Marvin Marcus, Di Xiong, Paula Ortega-Verdugo, Elizabeth Agredano, Yilan Huang, Linyu Zhou, Steve Y. Lee, Jie Shen, Ron D. Hays, James J. Crall, Honghu Liu

**Affiliations:** 1Division of Oral and Systemic Health Sciences, School of Dentistry, University of California Los Angeles, Los Angeles, CA 90095, USA; cmaida@ucla.edu (C.A.M.); mmarcus@dentistry.ucla.edu (M.M.); dixiong@ucla.edu (D.X.); eagredano@outlook.com (E.A.); yilanh19@ucla.edu (Y.H.); linyuzhou@ucla.edu (L.Z.); shenjie@ucla.edu (J.S.); jcrall@dentistry.ucla.edu (J.J.C.); 2Department of Biostatistics, Fielding School of Public Health, University of California Los Angeles, Los Angeles, CA 90095, USA; 3Division of Preventative and Restorative Sciences, School of Dentistry, University of California Los Angeles, Los Angeles, CA 90095, USA; portegaverd@ucla.edu (P.O.-V.); slee@dentistry.ucla.edu (S.Y.L.); 4Division of General Internal Medicine and Health Services Research, David Geffen School of Medicine, University of California Los Angeles, Los Angeles, CA 90095, USA; drhays@ucla.edu; 5Department of Health Policy and Management, Fielding School of Public Health, University of California Los Angeles, Los Angeles, CA 90095, USA; 6RAND Corporation, Santa Monica, CA 90407, USA

**Keywords:** focus group, patient-reported outcome measures, oral health, education, COVID-19, dental problem

## Abstract

This study reports the results of focus groups with school nurses and teachers from elementary, middle, and high schools to explore their perceptions of child and adolescent oral health. Participants included 14 school nurses and 15 teachers (83% female; 31% Hispanic; 21% White; 21% Asian; 14% African American; and 13% Others). Respondents were recruited from Los Angeles County schools and scheduled by school level for six one-hour focus groups using *Zoom*. Audio recordings were transcribed, reviewed, and saved with anonymization of speaker identities. *NVivo* software (QSR International, Melbourne, Australia) was used to facilitate content analysis and identify key themes. The nurses’ rate of “Oral Health Education” comments statistically exceeded that of teachers, while teachers had higher rates for “Parental Involvement” and “Mutual Perception” comments. “Need for Care” was perceived to be more prevalent in immigrants to the United States based on student behaviors and complaints. “Access to Care” was seen as primarily the nurses’ responsibilities. Strong relationships between community clinics and schools were viewed by some as integral to students achieving good oral health. The results suggest dimensions and questions important to item development for oral health surveys of children and parents to address screening, management, program assessment, and policy planning.

## 1. Introduction

The 2021 NIH Report on Oral Health provides a roadmap to improve the nation’s oral health, noting that dental caries is one of the most common diseases of childhood [1]; consequently, the report promotes oral health interventions and improved access to dental care for young children to promote oral health across the lifespan. Patient-reported outcomes (PROs) can be used to assess oral health and evaluate the impact of dental care [2]. The Patient-Reported Outcomes Measurement Information System (*PROMIS^®^*) includes reliable, valid, and efficient patient-reported measures for a wide range of chronic diseases and demographic characteristics to evaluate medical interventions [3,4]. However, *PROMIS^®^* item banks, as tools to assess outcomes, are just beginning to target children’s oral health.

Qualitative methods include focus groups and interviews [5,6], which emphasize “grounded” concepts and theories [7]. To achieve a grounded approach, we used nurses and teachers as knowledgeable informants [6,8].

Oral health education can be integrated into the school routine, promoting students’ knowledge of oral health conditions and oral hygiene habits [9,10]. Teachers and school nurses have unique perspectives about oral health education, promotion of student oral health, and oral health literacy of parents and caregivers. With increasing frequency, teachers and nurses provide education programs to improve children’s understanding of their oral conditions and good dental habits [11,12,13]. Kwan et al. provide a comprehensive perspective on the consequences of the shortage of dental personnel trained in oral health promotion, citing examples of the inability of teaching staff members, not trained in this area, to assist in carrying out these essential efforts [14]. To foster these efforts, teachers’ oral-health-related knowledge, attitudes, and practices need to be strengthened [15,16,17]. Continuing education and in-service training are critical for school nurses who, as the health professionals in their schools, play an essential role in oral health promotion [18,19,20].

Perceptions and practices of teachers and school nurses are changing due to available dental resources, advanced technologies, and remote teaching, especially during the COVID-19 pandemic [21,22]. A survey of pediatric dentists in Germany found that only one-fifth of dental clinics were fully functional during the first lockdown, with two-thirds of respondents reporting that oral hygiene appointments and regular dental checkups were postponed [23]. Remote consultations and digital photographs were used as an alternative way to screen patients by the pediatric dental emergency service to minimize unnecessary emergency visits [22,24].

Traditional focus group discussions involve face-to-face interactive group settings and usually experience barriers such as participant dropout and financial costs. There is increased attention toward utilizing online environments to facilitate aspects of focus group discussions. More and more studies have employed an entirely virtual process to focus group research, from recruitment to facilitation and reimbursement [25,26,27,28].

The purpose of this qualitative research study is to elicit nurses’ and teachers’ perceptions of school-aged children and adolescent oral health status and reflect on their experiences with students and parents.

## 2. Materials and Methods

### 2.1. Recruitment of Participants

Nurses and teachers were recruited from three school levels including elementary, middle, and high schools across Los Angeles County. The school sites were selected across all eight service planning areas (SPAs) [29]. Participants were contacted between May and July 2021 via multiple sources, such as previous school connections, email groups, and referrals. A flyer with a description of the study objective, procedure, and incentive was emailed to the potential participants. The potential participants completed a HIPAA-compliant *Google Form* (Google LLC., Mountain View, CA, USA) to provide information, such as school name(s), school district, grade, gender, and race/ethnicity. All participants were required to speak English and currently work in at least one school site. The participants received a $50 *Amazon* e-gift card upon finishing the interviews. All subjects gave their informed consent for inclusion before they participated in the study. The study was conducted in accordance with the Declaration of Helsinki, and the protocol was approved by the Ethics Committee of the Office of Human Research Protection Program, University of California, Los Angeles (Project identification code 20-000719). The research team has published previous studies on children’s oral health [30,31,32]. The work reported here is presented according to the recommendations of the Standards for Reporting Qualitative Research (SRQR), as listed in Appendix A [33].

### 2.2. Focus Group Interviews

We conducted and recorded three one-hour focus group interviews for teachers at three school levels separately, and another three for school nurses, via *Zoom*, a HIPAA-compliant online meeting platform [34]. Each meeting was scheduled for a time in the late afternoon or early evening from June to July 2021 and included three-to-six participants in each session, with an invitation and multiple reminders sent to six-to-ten potential participants. The discussions were guided by a moderator, an observer, and at least one assistant moderator. Interviews were performed based on a two-stage framework with a schedule of questions drafted by two experts in the field of children’s oral health, covering the following: students’ need for and access to care; referrals to dental providers or community clinics; oral health education; parental involvement; mutual perceptions of teachers and nurses; and the use of remote technology in teaching and school nursing practice. In the first stage, the moderator led an open discussion for each question around participants’ experiences with students’ oral health. The second stage further allowed participants to delve deeper into specific areas, followed up by questions posed by the observer at the end of the discussion. The questions, sub-questions, and prompts guiding the discussion are shown in Appendix B.

Before starting the recording of each session, participants provided their audio consents and were given a participant ID to assure anonymity during the whole session. Participants’ focus group comments and background information were linked using the assigned participant ID. Data collection included audio recordings, focus group transcripts, and moderator comments. To assure the confidentiality and protection of participants, audiotaping and audio recordings were only accessed by the research team.

### 2.3. Analysis

Each focus group was digitally recorded and transcribed verbatim and entered into *NVivo* software (Released in March 2020 by QSR International, Melbourne, Australia), a qualitative text analysis database [35]. Frequencies and content analysis of the focus group transcript, including thematic and narrative analyses, were used to uncover themes. The frequency of mentions pertaining to specific themes, for example, provided a sense of what nurses and teachers viewed as important. We reviewed the constructs within each focus group; reviewed the total data set to identify promising thematic domains; grouped responses according to these thematic domains; and then finalized a set of six key themes. We examined how these themes supported the data and our primary theoretical perspective, namely students’ oral health status, and then determined the frequency of mentions, or expressed concerns, for each theme by occupational role. The comparison between school nurses and teachers has been compared using a chi-square test with a significance level of 0.05 [36]. The broad issues explored in the discussions were classified as “coding nodes” to document the frequency of expressed concerns related to a particular theme or topic and to analyze relationships between these concerns based on the respondents. Within each coding node, specific content was coded as pertinent to more general and/or more particular issues to allow for multi-level analysis. 

## 3. Results

### 3.1. Participants

Six focus groups were conducted: three for the school nurses (*N* = 14) and three for teachers (*N* = 15). The self-reported demographic information is summarized in Table 1. Most participants were female, with 100% of school nurses and 67% of teachers identifying as female. In addition to three participants who preferred not to disclose their race and ethnicity, over one-third of school nurses were Hispanic (36%), with 21% identifying as White and another 21% as Asian. Most of the teachers were Hispanic (27%) and African American (27%). To be more representative, we recruited participants from different school types, including eight private, seven public general, five public magnet, and four public charter schools. Since some of the schools shared one school nurse, another four district nurses were interviewed. More details about participants from different school types are presented in Table 1. 

### 3.2. School Nurse and Teacher Responses

Overarching issues included: (1) students’ needs for oral health treatment and access to needed services; (2) concern for the oral health needs of “newcomers”, namely students of recent U.S. immigrant and refugee families; and (3) challenges to parents’ involvement in their children’s oral health concerns while advancing various approaches to enhance parental engagement in attending to them. 

Table 2 provides data on the distribution of nurses’ and teachers’ comments for the six key themes: need for care, access to care, referrals, oral health education, parental involvement, and mutual perceptions of nurses’ and teachers’ roles in students’ oral health concerns. Each theme consists of positive and negative codes. This table presents (1) the frequency of comments related to each theme and the number of comments per nurse or teacher; and (2) the number of nurses or teachers who made one or more comments containing the thematic code, and the proportion of respondents within the group.

The most frequent comments by nurses concerned “Access to Care” (*n* = 35) and the least frequent responses were two comments on “Mutual Perceptions”, made by nurses about teachers and vice versa, while the most frequent comments made by teachers concerned “Parent Involvement” (*n* = 56), and the least frequent concerned “Referrals” (*n* = 8). There was a statistically significant difference (*p*-value < 0.01) between nurses and teachers regarding the number of comments on “Oral Health Education” (rates 1.71 vs. 0.67), “Parental Involvement” (rates 2.07 vs. 3.73), and “Mutual Perceptions” (rates 2.07 vs. 3.73). Most nurses responded to all except for “Mutual Perceptions”, which was mentioned by only two respondents. Teachers also responded at relatively high rates except for “Referrals” (rate = 0.47) and “Oral Health Education” (rate = 0.27). “Mutual Perceptions” was the only theme that was statistically significant (*p*-value = 0.02). Table 3 provides quotes to illustrate how these themes are expressed in the teachers’ and nurses’ own words. All quotations cited in the text are those of focus group participants.

**Need for Care**—Teachers noted how students moved from being embarrassed about their teeth, appearing reticent about smiling, purposely covering their faces, and keeping their mouths tightly closed, to becoming open and smiling once they received dental treatment, including braces. Teachers expressed concern for the many students from new immigrant families, especially from Central and South America where there were limited resources for the poor, stating that, once they came to this country, it was often the first time that these students encountered a dental professional. Nurses acknowledged the same, claiming that many newcomers present with other physical disabilities that will often place oral health treatment at a lower priority for parents. Often frustrated in their efforts to influence a parent to follow up on their referrals for a child with oral pain, some nurses employed fear arousal, stating that left untreated, infections in the oral cavity may place their children at risk for other physical ailments.

**Access to Care**—Teachers stated that they are typically the first to recognize a dental problem and send the student to the nursing office. Sometimes, a child will have recurrent dental problems, especially those from specific populations, including the child welfare and juvenile justice systems, where there may be limited follow up by guardians or family caregivers. In such cases, teachers enlisted social services to assure access to needed treatment. Nurses spoke about assuring access, even for immigrant or refugee students, by advocating for parents to register their children for Medicaid services. Even with this push for increased access to dental care, nurses acknowledged the need to increase parents’ oral health literacy, conveying the importance of seeking immediate care for untreated conditions that may affect their child’s learning. Teachers and nurses agreed that students are more likely to have poor oral health if they are members of low-income families, uninsured, members of a racial or ethnic minority, or are from immigrant families.

**Referrals**—Teachers expressed gratitude for the availability of low-cost dental resources in community and local dental school clinics. Despite this, many addressed concerns about students from “newcomer” families, those in group homes, or the juvenile justice system, where there were many barriers to scheduling and keeping appointments once a referral was made. Nurses viewed the contractual arrangements between local dental schools and third party in-school dental programs as facilitating their referrals. Most had listings of low-cost community resources to refer students for needed treatment and found that parents were receptive to the dental staff’s recommendations and satisfied with the care their children received.

**Oral Health Education**—Teachers rarely viewed themselves as health educators, viewing oral hygiene and prevention as best left to dentists and clinics with contractual agreements with their schools. Some teachers were grateful for the clinic representatives that present to students, across the grades, on daily oral hygiene and regular dental care. Nurses talked about their limited formal training in oral care when in nursing school, though some expressed how they learned from both teachers who taught health classes and online courses. Some nurses have even learned sufficient content to work with teachers and lead in instructing the oral hygiene class.

**Parental Involvement**—Teachers expressed challenges in communicating in the various languages, including indigenous dialects, prevalent in their school communities. Despite such language barriers, they recognized the importance and willingness of many non-English-speaking parents to participate in their children’s learning. Other teachers pointed to potential barriers that limit parental involvement, notably fears about disclosure of the families’ legal status, and defensiveness about their long working hours that hinder the ability to fully participate in their child’s schooling. Teachers appeared positive about the use of remote technology platforms to engage parents and facilitate communication, by visually sharing what their children are learning in class. While school nurses have used remote technology in reaching parents, they preferred face-to-face parent meetings to share resources, suggesting working with parent centers and their representatives as the best way to increase such interactions. Nurses were concerned that certain parents do not respond to their calls regarding the child’s pain, noting that, even when they do call back, some appeared unable to handle the matter themselves, viewing the nurse as the expert best suited to manage their child’s health needs.

**Mutual Perceptions**—The teachers’ comments regarding nurses represented extremes, ranging from praise to criticism of their performance. While some teachers expressed concerns about part-time nurses’ availability, others recognized the school nurse’s ability to interact with parents and serve as a resource for referral to community services. Nurses regarded teachers as an effective resource to alert them to students’ oral health concerns because of their ongoing contact with students. While some nurses said that the inability to provide oral hygiene supplies to classrooms precluded their engagement with teachers in an oral health activity, others worked with teachers to incorporate oral health as part of required health classes.

## 4. Discussion

School nurses and teachers from elementary, middle, and high schools amid the COVID-19 pandemic conveyed their insights into students’ oral health and the challenges these professionals face while working in diverse school communities. All commented about challenges with involving parents in their children’s educational and health concerns, although teachers expressed almost twice as many comments as nurses. The pandemic necessitated their interaction with parents, though parents’ suboptimal responses exacerbated their frustrations, sometimes to the point of alienation. Teachers had much to say about the nurses’ involvements with their students. Most comments were laudatory; however, some teachers expressed frustration with nurses’ lack of attention to their students’ oral health needs. Other teachers mentioned that their schools did not even employ nurses. Nurses also expressed frustration, notably that the considerable time necessary to conduct required hearing and eye examinations impinged upon their ability to address students’ oral health problems. Nurses were more involved in oral health education than teachers, while neither group felt sufficiently trained to conduct such programs.

Focus group participants noted that the pandemic required using remote technology to teach, educate, evaluate, and communicate with parents and students. Nursing’s traditional role of providing face-to-face contact with students and parents was conducted in a medium that required separation, as remote technologies were commonly relied upon. Teachers often competed with nurses to remotely access children and their parents. Parents raising families in crowded homes, with uncertain language skills, and multiple jobs, may feel impinged upon, while other parents may welcome this modality because of the convenience of not having to travel to school to meet with teachers or nurses. When the pandemic subsides and as children return to school, the technology’s impact is expected to alter in overt or subtle ways the roles of everyone engaged in the educational process and, based upon lessons learned, may enhance the presence of oral health.

The need for effective oral health interventions is particularly important because it links dental schools to K-12 educational settings in partnerships that can provide educational opportunities for dental student involvement in community-based care. The underlying sense is that there is a gap between what is being carried out in schools to enhance oral health programs and referral networks for needed services through engaging nurses and teachers in oral health education and advocacy, and the establishment of broader-based coalitions on behalf of students’ oral health needs. Such partnerships between the K-12 schools, dental schools, and dental providers in the community are key to assuring targeted advocacy efforts for access to needed services. In Los Angeles, a nonprofit organization used key informants and focus groups to organize and activate a community coalition to fund a district-wide oral health nurse program to implement a school-based intervention with an array of services including screenings, fluoride applications, and referrals [37]. Another effort advocated for school nurses to promote the oral health of children and families, including specific actions and oral health information [38].

There are certain benefits to remote focus groups. First, this modality saves time and money associated with travel for both the researcher and the participant within these virtual settings [26]. Second, the remote techniques allow simultaneous participation across a wide geographic area, providing the potential for geographically diverse participants [1]. Third, remote focus groups provide an anonymous and comfortable environment to assess sensitive experiences, particularly for marginalized populations who may be unwilling to participate in face-to-face focus groups [27]. At the same time, remote focus groups have the disadvantages of the lack of face-to-face interaction, connectivity issues, and technical missteps. Besides, it is difficult for some participants to find a quiet and private place with minimal disruptions [28]. Beyond these generalized limitations, our study is limited to a small group of teachers and nurses who responded to online recruitment efforts and felt comfortable participating in a video-recorded online discussion. While participants provided valuable responses and a range of opinions that will guide oral health item development for our survey research, they clearly did not represent their many colleagues in schools across the region.

Findings from the focus group discussions with nurses and teachers nevertheless showed the critical importance of developing a strong relationship between the schools and community dental clinics, which parents rely on to provide their children’s treatment. Locating reliable resources for proper access to care and finding dentists to care for the children were two of the main discussion topics during our sessions. An underutilized and less recognized resource for the healthcare delivery model is the dental school, which can offer students and residents as valuable partners in community care. A study demonstrated how a dental school-operated community clinic using dental students to provide care can be successfully implemented and meet the needs of the community, which highlighted how such a program can enhance the education and training for students and residents while meeting the needs of an underserved community [39,40]. A more recent model, Community-Based Clinical Education (CBCE), adopted by over a quarter of American dental schools, increased the access to care for local areas while enhancing education and training for the students and residents, utilizing a financially sustainable model [40,41]. Additionally, the early exposure of the dental students and residents to public service provided the opportunity for graduates to consider seeking employment in community public health systems [41,42].

Hence, establishing a collaboration between schools, community dental clinics, and dental schools is critical to improving the overall health and wellbeing of children and the school community.

## 5. Conclusions

This study provides insights on the perceptions of child and adolescent oral health and practices of school-based professionals during the COVID-19 pandemic, which disrupted their contact with students and their parents. Employing remote technology, teachers and nurses assessed the advantages and challenges of this communication modality. All participants expressed a need for additional oral health training to equip them to be effective in their roles. Both teachers and nurses valued access to dental schools and community clinics as they set out to navigate their students’ needs for dental care. The results suggest dimensions and questions important to item development for oral health surveys of children, adolescents, and parents to address screening, management, program assessment, and policy planning.

## Figures and Tables

**Table 1 ijerph-19-04722-t001:** Characteristics of nurses and teachers by session.

Characteristics	Nurses: N = 14 *n* (%)				Teachers: N = 15 *n* (%)				Total
	Elementary School	Middle/High School ^1^	High School	Total	Elementary School	Middle School	High School	Total	
**No. of Participants**	6 (100%)	5 (100%)	3 (100%)	14 (100%)	5 (100%)	5 (100%)	5 (100%)	15 (100%)	29 (100%)
**Gender**									
Female	6 (100%)	5 (100%)	3 (100%)	14 (100%)	4 (80%)	3 (60%)	3 (60%)	10 (66.7%)	24 (82.8%)
Male	0 (0%)	0 (0%)	0 (0%)	0 (0%)	1 (20%)	2 (40%)	2 (40%)	5 (33.3%)	5 (17.2%)
**Race/Ethnicity**									
Hispanic/ Latinx	2 (33.3%)	3 (60%)	0 (0%)	5 (35.7%)	2 (40%)	1 (20%)	1 (20%)	4 (26.7%)	9 (31.0%)
White/ Caucasian	2 (33.3%)	1 (20%)	0 (0%)	3 (21.4%)	1 (20%)	0 (0%)	2 (40%)	3 (20%)	6 (20.7%)
Asian/Asian American	1 (16.7%)	1 (20%)	1 (33.3%)	3 (21.4%)	1 (20%)	1 (20%)	1 (20%)	3 (20%)	6 (20.7%)
Black/African American	0 (0%)	0 (0%)	0 (0%)	0 (0%)	1 (20%)	2 (40%)	1 (20%)	4 (26.7%)	4 (13.8%)
American Indian/Alaskan Native	0 (0%)	0 (0%)	0 (0%)	0 (0%)	0 (0%)	1 (20%)	0 (0%)	1 (6.7%)	1 (3.4%)
Prefer not to say	1 (16.7%)	0 (0%)	2 (66.7%)	3 (21.4%)	0 (0%)	0 (0%)	0 (0%)	0 (0%)	3 (10.3%)
**School Type**									
Private School	1 (16.7%)	3 (60%)	0 (0%)	4 (28.6%)	1 (20%)	3 (60%)	0 (0%)	4 (26.7%)	8 (27.6%)
Public General School	2 (33.3%)	0 (0%)	0 (0%)	2 (14.3%)	2 (40%)	1 (20%)	1 (20%)	4 (26.7%)	6 (20.7%)
Public Magnet School	0 (0%)	0 (0%)	2 (66.7%)	2 (14.3%)	0 (0%)	1 (20%)	3 (60%)	4 (26.7%)	6 (20.7%)
Public Charter School	0 (0%)	1 (20%)	1 (33.3%)	2 (14.3%)	2 (40%)	0 (0%)	1 (20%)	3 (20%)	5 (17.2%)
District ^2^	3 (50%)	1 (20%)	0 (0%)	4 (28.6%)	0 (0%)	0 (0%)	0 (0%)	0 (0%)	4 (13.8%)

^1^ A mixed session with nurses from both middle and high schools. ^2^ Nurses can work at multiple school sites at different education levels in one district. They participated in only one session.

**Table 2 ijerph-19-04722-t002:** Frequency of mentions, by teacher and nurse respondents, for the key themes.

Themes	Frequency of Comments					Frequency by Respondents				
	School Nurses (N = 14)		Teachers (N = 15)		*p*-Value ^3^	School Nurses (N = 14)		Teachers (N = 15)		*p*-Value ^3^
	*n*	Rate ^1^	*n*	Rate ^1^		*n*	Rate ^2^	*n*	Rate ^2^	
Need for Care	25	1.79	33	2.20	NS	10	0.71	13	0.87	NS
Access to Care	36	2.57	25	1.67	NS	11	0.79	13	0.87	NS
Referrals	14	1.00	8	0.53	NS	12	0.86	7	0.47	NS
Oral Health Education	24	1.71	10	0.67	<0.01 **	9	0.64	4	0.27	NS
Parental Involvement	29	2.07	56	3.73	<0.01 **	13	0.93	15	1.00	NS
Mutual Perceptions	2	0.14	14	0.93	<0.01 **	2	0.14	11	0.73	0.02 *

^1^ Rate—number of comments/total number of participants; ^2^ Rate—number of respondents/total number of participants; ^3^—chi-square test; NS—not significant; *: *p*-value < 0.05; ** *p*-value < 0.01.

**Table 3 ijerph-19-04722-t003:** Representative comments for key themes.

Themes	Representative Comments	
	Nurses	Teachers
**Need for Care**	**Elementary School Nurse**: When they come in with mouth pain, I have them rinse their mouth. I do have them floss...I also tell them that you know they’re going to end up getting an infection and it’s going to go to their heart. **Middle School Nurse**: But a lot of these children are coming from other countries where they have other physical impairments, orthopedic impairments, things like that and oral health is not a priority for them…There’s some shame also, I think, that comes along with that, where they won’t open their mouth, or they won’t smile only because they know that they are lacking oral hygiene and care.	**High School Teacher**: So, I’ve known students who, bay don’t s and when they laugh, they try to purposely cover-up, so they’ll keep their mouth closed. But then, from students I’ve known, they’ll get braces and, afterward, once they’re done with the braces, suddenly, they start smiling again. So, and I talked about this before, about how they were embarrassed about their teeth and, you know, until they got the braces and their teeth were straightened; now, then, afterward, they smile differently. **High School Teacher**: With a lot of our newcomers that are coming from El Salvador, Guatemala, Nicaragua, Colombia, Mexico, especially from Central and South America, many of them are coming. with zero dental oral health services in their lives, and so it, you know, especially that population, has struggled and then, you know, come here and then really needs a full workup analysis, I don’t know what dentists call it, but, you know, some indication.
**Access to Care**	**Elementary School Nurse**: Well, if they want to get the care, they will. What happens is they cannot get the care? Then, the group that I refer them to makes sure that they help them register for Medi-Cal. I make it quite clear with the parent; if they are an immigrant, it makes no difference if they are a citizen or not. Your child is eligible to have care. It doesn’t mean that you are eligible to have care, but your child is eligible to have care, so please, please, your child is in need of care and that will help them do better in school. **Middle School Nurse**: We do have access to a lot of services; the mobile services are limited, varnish, fluoride treatments, screening and then, if there are severe diseases or things that can’t be done in a mobile setting then we refer them out. Part of the issue is a lot of it is also, for at least our demographic, a lot of education that’s needed around oral health sometimes. The family is aware that the child has oral health needs, but just doesn’t understand the importance of dental care and what could potentially happen if they don’t get that treated immediately. And the other thing is, we do have a lot who are fearful of whether it’s their immigration status or their financial status or if their Medi-Cal or Denti-Cal. There’s that fear that they will still be financially responsible so they’re not willing to seek the resources out of fear, so we find that we have to do a lot of education with our students and our families.	**Middle School Teacher**: The health clerk checks them out and usually contacts parents and if they don’t get ahold of parents, I usually like to follow up, especially with my special population because that’s the experience I’ve had that they’re the same kids that are constantly having the same issues. The first one, that I had in mind, you know, he was always saying you know, he was always hungry and always complaining of the toothache, so we had to call the guardians more than once to be able to reach them. But sadly, in his case, we had to get social services involved because there was clear neglect with him, unfortunately. **High School Teacher**: Yes, in my previous school that I was at for eight years, the majority of my students in my program were from the group home and our justice system and so forth, so there were lots of issues with consistency as far as their oral hygiene was concerned So, certainly, there is the aspect that the students were in pain from dental issues and the group home did make phone calls and appointments for the students to see a dentist.
**Referrals**	**Elementary School Nurse**: In my district, we use a dental school and also use an in-school dental program and they come to this school but, like you said two-three days only; they see students in three sessions; they wait in lines to be seen. They can do X-rays, fluoride treatments, and those who don’t have insurance, receive only cleaning and they put the fluoride. They tell the parents to take their child to the doctor at the dental school who does the treatment…So, I refer them to the ones that cover Medi-Cal, because the majority of these kids have Medi-Cal. So, I refer them to the closest one where they live. Sometimes when I screen, I scare them. They have a lot of scary mouths in my area, so. I refer them to the doctor, who sometimes asks why the child has not been a dentist. **Middle School Nurse**: We do see a lot of students with caries or even ones that need braces and don’t have access to it, so in those instances, first of all, make contact with the family, with a parent; see what kind of insurance or lack of insurance they have; what kind of access. If they need resources, we’ll be for them; we’re fortunate that we have some school-based clinics. We do have mostly volunteers who work, and then we also have mobile oral health programs that visit our school sites or that we can refer students to.	**Middle School Teacher**: We don’t have a school nurse, but because we’re a community school, we have a lot of different referrals. So, we were able to refer them out to different programs. We also work closely with a local clinic, which is a free clinic and then in our community, which is there for them as well and we’re close to a dental school, so we usually refer them to those two places. **High School Teacher**: The kids who are in group homes, the kids who are part of the juvenile justice system; those kids often struggle with oral health issues and oftentimes cannot get dental appointments. Part of the reason why, you know, we set up a partnership with a dental school, so that that we could get some dentists to come out and try to provide at least some screening and what not for some of our students who are not yet at least, in our case, in the group homes and the juvenile justice system, which are not necessarily making the appointments for them. And so, sometimes these kids are, you know, they’ll come, and they’ll say, you know, I’m in pain and you know, and then we’ll try to hook them up with a dentist or at least, you know, a local clinic that can at least provide some level of support.
**Oral Health Education**	**Middle School Nurse**: I work with the main teacher that teaches that health course who provided kind of the material to me, and then we added to that with online resources and things like that, and then formal training I’d say nursing school does touch on oral hygiene, especially in pediatrics rotation and then I can say for myself my community health rotation was at a free clinic where we actually worked alongside dentists, and I did dental hygiene work as a nursing student as well, so in that sense, I got to, you know, kind of hopefully work in a space with a dentist and be in that same space. **Middle School Nurse**: I didn’t have any formal training, but I have a good relationship with one of the parents there who happens to be a dentist. So, any questions I have, I always refer to him, and then, when I do teach a health class, it is material that has been passed down to me from the previous teacher and I usually just focus on dental hygiene, the, you know, the important part of it, the name of the teeth and washing and why it’s important but also what I see when the kids come in with any dental problems I’ll have an example about that, and why it got to that point and what to do after.	**Elementary School Teacher**: We don’t really teach oral hygiene and oral health is for dentists. I think it’s left up to the parents and my experience in the past has been that schools have been involved with bringing a dentist in giving them a personal lesson on what to do, how to encourage them to know how to brush their teeth after eating, making sure that that’s done. **High School Teacher**: We are lucky enough in my area to have a clinic, that is, you know, a very, very generous clinic that helps our students and this clinic, actually, they present to our students through elementary school, and they try to get to middle school and high school. So, you know, I even have this group come in and present too, so we talked a lot about, you know, dental care and basic hygiene and how important it is for our health in general that, you know, you could have numerous health problems, just because your teeth are not, you know, healthy.
**Parental Involvement**	**Elementary School Nurse**: Another nurse came up with an amazing plan, where she would set up a table in the morning with the puppets and with the different dental materials. Then, as parents were dropping off kids, that’s another way to get the parents to sign consents because the afterschool program doesn’t reach the parents that are the stay-at-home moms that do not have their kids in after-school programs. Also, present visuals that everyone would just be curious about: the cute dinosaurs and also have giveaways at that table that you give out. It’s not even about toothbrushes you give out. They did the best job of engaging parents that I had ever seen. **High School Nurse**: A lot of the schools have an active place where parents can gather and share resources and, on my campus, we have a parent representative who oversees the parent center and interacts with parents. So, you asked how we can get the parents involved: create awareness, be invested in or oversee, you know, supervising their children’s health. I would say that’s a really good avenue to tap into and if there are agencies for programs where they want to provide, you know, education, like seminars, a demo of good oral hygiene, health and how to do that. I mean, even at the high school level, you know high schoolers are lazy. You know and unless it’s drilled in, something that they do on their own, they’re going to neglect that so and then, if I mean oral hygiene, I think, is always tied into nutrition so. You know, combining that with your basics good nutritious eating and at the high school well that’s a problem because kids don’t eat and they don’t choose nutritious foods and so that ties back into getting, you know, caries and all that stuff so.	**Elementary School Teacher**: I’d keep it small only because you’ll have parents that maybe don’t feel like they can chat; maybe their English isn’t strong, if you know, maybe they didn’t get it; they might not even be speaking Spanish, they might speak another language...We see this with a lot of second language families, where they’re so capable, but because they’re not strong in English, they don’t volunteer to help out at the school. And they can give so much to our school, so you’ll see this also where they won’t share; so maybe have something after where they can, I don’t know, where they can talk some, you know e-mail or do something that they can continue after, because a lot of them will just sit there and smile, but they have a lot to say; you just won’t know it, we see that a lot. **Elementary School Teacher**: I would just say you’re going to get a higher rate of attendance on Zoom; that’s what I’ve noticed, especially with conferences. For parents who have a hard time at lunch break to drive all the way down to school, Zoom has made it so much easier to communicate. So, I think people are going to be, especially parents are going to be, much more open to doing a meeting knowing they don’t need to drive somewhere, find parking and find a babysitter and such.
**Mutual Perceptions**	**Elementary School Nurse**: You know, I have tried to teach some teachers. The thing is we lack supplies, like toothbrushes and then just to actually teach the kids, because that’s what I had mentioned. I want to do a group of my teachers at my school. I remember back in the day, they used to be like when everyone came back from lunch, they used to say, “brush your teeth.” Just teaching those healthy habits at a young age. I tried to do that, but it was just a lack of supplies, lack of resources, so it hasn’t been done, but teachers are receptive. But it’s not something you know they can afford. **Middle School Nurse**: As with other health issues, the teacher always is the first one to figure out something is wrong because they’re the ones with students almost every single minute at school. We are like, “whenever they find the problems… well, send them to us”, so our teachers will definitely help a lot.	**High School Teacher**: Our school nurse is absolutely fabulous. She is most of the time at our school. She’s the head nurse for the district, so she spends a lot of time at our school, but there are health assistants. She has health assistants and then she also has nurses in training that come in and they are fabulous; they are wonderful; they’re very receptive to any issues that we have on campus. So, I definitely wouldn’t have any problem referring a student down to the nurse, but I also wouldn’t have any problem referring a student to our, you know, clinic in the area that could help out. **High School Teacher**: We do have a full-time school nurse, and I have the absolute opposite experience. She often sends kids back immediately and won’t even call home if they’re sick. They pretty much go to her with whatever their ailment is, and she gives them an ice pack and sends the student back to class. That’s correct, and then to give her a little credit, we do have a high population of students with significant disabilities, so she does, you know, help supervise with toileting for some of the kids. Or some of our students receive medication through the day and things like that, but overall, just the general student coming in with any kind of problem, they’re not going to get any help through the nurse’s office. They are more likely to get help if we send them to their academic counselor.

## Data Availability

The data presented in this study are available on request from the corresponding author. The data are not publicly available due to research participants’ privacy.

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
