# Peer review of "Investigating Perceptions of Teachers and School Nurses on Child and Adolescent Oral Health in Los Angeles County"

_ijerph, 2022, doi:10.3390/ijerph19084722_

Round 1

Reviewer 1 Report

The manuscript contains examination of  children and adolescence oral health perception in a group of school nurses and teachers. The study is well planned and results are presented in very detailed manner. 

If it would be possible to summarize the Table number 3 in the way it would be more clear for the readers, maybe  show the key words or key sentences instead of whole statements of participants.

Typical for the scientific manuscripts are data clustering, not extensive description.

Author Response

Point 1: If it would be possible to summarize Table number 3 in a way it would be more clear for the readers, maybe show the keywords or key sentences instead of whole statements of participants. Typical for the scientific manuscripts are data clustering, not extensive description.

Response 1: As we set about to edit Table 3 in response to the reviewer’s comments, it became apparent to us that any further editing would lose the essence of the individual nurses’ and teachers’ responses. The extensive description is a necessary part of qualitative research, as we have put forth in this paper based on focus group methodology. 

Reviewer 2 Report

1. Perhaps the title can be refined to: Investigating Perceptions of Teachers and School Nurses on Child and Adolescent Oral Health in Los Angeles County

2. A useful and applicable reference pertaining to interprofessional collaboration to strengthen the discussion in lines 280-293: Modha, B.(2022), "Utilising dentist-dental health educator skill-mix to implement oral health promotion that better supports diverse communities", Journal of Integrated Care, Vol. ahead-of-print No. ahead-of-print. https://doi.org/10.1108/JICA-08-2021-0043 

3. A useful and relevant reference that considers school-based oral health education that may enhance the discussion in lines 309-325: Nguyen, V.T.N., Zaitsu, T., Oshiro, A., Tran, T.T., Nguyen, Y.H.T., Kawaguchi, Y. and Aida, J., 2021. Impact of School-Based Oral Health Education on Vietnamese Adolescents: A 6-Month Study. International journal of environmental research and public health18(5), p.2715.

4. Overall, a well written, articulate paper that covers a very relevant and important issue that shall be of benefit to an international audience. The figures, references, and statistics are of satisfactory standard. The above suggested references would make valuable additions.

5. Please ensure the language, grammar, punctuation, spelling and sentence structures are carefully assessed and refined to ensure a succinct and coherent read. This is because there are minor discrepancies in the language, grammar, punctuation, spelling and sentence structures. If need be, please obtain the necessary scientific English language reading and editing assistance, so that the paper has the potential to be read enjoyably by the international readership.

Author Response

Point 1: Perhaps the title can be refined to, "Investigating Perceptions of Teachers and School Nurses on Child and Adolescent Oral Health in Los Angeles County".

Response 1: We have changed the title of this manuscript to accord with the reviewer’s suggestion.

Point 2: A useful and applicable reference pertaining to interprofessional collaboration to strengthen the discussion in lines 280-293: Modha, B.(2022), "Utilising dentist-dental health educator skill-mix to implement oral health promotion that better supports diverse communities", Journal of Integrated Care, Vol. ahead-of-print No. ahead-of-print. https://doi.org/10.1108/JICA-08-2021-0043 

Response 2: The paper suggested by Reviewer 2 is an excellent example of a school-based oral health education program.  However, the focus of our paper is on the role of teachers and school nurses, rather than that of dental personnel. We have reviewed the paper and found that this study is conducted by dentists. The authors state the following: “The education session started with a 15-min power-point lecture conducted by a dentist in the classroom. The lecture focused on dental plaque formation and characteristics, gingivitis etiology and symptoms, toothbrushing technique, and the role of a sugary diet in dental caries and gingivitis. After completing the lecture, all students observed their mouths with a hand mirror. Students recorded the dental plaque and gingivitis condition of their anterior teeth on the self-assessment form. They subsequently used a disclosing solution… to see dental plaque on their teeth. Students recorded their plaque condition again and compared it with the previous one. After this, each participant received a toothbrush to brush their teeth while using the hand mirror to observe their brushing efficacy. In the hands-on session, the students performed all the activities independently without the educator’s supervision. The dental mirror and toothbrush were given to participants to practice at home, and no further education was provided within the follow-up.”

Point 3: A useful and relevant reference that considers school-based oral health education that may enhance the discussion in lines 309-325: Nguyen, V.T.N., Zaitsu, T., Oshiro, A., Tran, T.T., Nguyen, Y.H.T., Kawaguchi, Y. and Aida, J., 2021. Impact of School-Based Oral Health Education on Vietnamese Adolescents: A 6-Month Study. International journal of environmental research and public health18(5), p.2715.

Response 3: Rather than adding the above two papers to the background, we believe that Kwan et al, “Health-Promoting Schools” addresses the key issues of our paper.  This will also provide a global health view of oral health education for an international readership, per another one of the reviewer’s suggestions. In the background, we discuss and cite the following reference:

Kwan, S. Y., Petersen, P. E., Pine, C. M., & Borutta, A. (2005). Health-Promoting Schools: An Opportunity for Oral Health Promotion. Bulletin of the World Health Organization 2005 83, 677-685.

Added to the text: “Kwan et al. provide a comprehensive perspective on the consequences of the shortage of trained dental personnel in oral health promotion, citing examples of the inability of teaching staff members, untrained in this area, to assist in carrying out these essential efforts.”

Point 5: Please ensure the language, grammar, punctuation, spelling and sentence structures are carefully assessed and refined to ensure a succinct and coherent read. This is because there are minor discrepancies in the language, grammar, punctuation, spelling and sentence structures. If need be, please obtain the necessary scientific English language reading and editing assistance, so that the paper has the potential to be read enjoyably by the international readership.

Response 5: We have performed a grammar and spelling check of the entire manuscript to ensure that the paper represents the necessary scientific English language for an international readership.